# Dichotomy of Control: Separating What You Can Control from What You Cannot

**Mengjiao Yang**
University of California, Berkeley
Google Research, Brain Team
`sherryy@berkeley.edu`

**Dale Schuurmans**
University of Alberta
Google Research, Brain Team

**Pieter Abbeel**
University of California, Berkeley

**Ofir Nachum**
Google Research, Brain Team

## Abstract

Future- or return-conditioned supervised learning is an emerging paradigm for offline reinforcement learning (RL), where the future outcome (i.e., return) associated with an observed action sequence is used as input to a policy trained to imitate those same actions. While return-conditioning is at the heart of popular algorithms such as decision transformer (DT), these methods tend to perform poorly in highly stochastic environments, where an occasional high return can arise from randomness in the environment rather than the actions themselves. Such situations can lead to a learned policy that is *inconsistent* with its conditioning inputs; i.e., using the policy to act in the environment, when conditioning on a specific desired return, leads to a distribution of real returns that is wildly different than desired. In this work, we propose the *dichotomy of control* (DoC), a future-conditioned supervised learning framework that separates mechanisms within a policy's control (actions) from those beyond a policy's control (environment stochasticity). We achieve this separation by conditioning the policy on a latent variable representation of the future, and designing a mutual information constraint that removes any information from the latent variable associated with randomness in the environment. Theoretically, we show that DoC yields policies that are *consistent* with their conditioning inputs, ensuring that conditioning a learned policy on a desired high-return future outcome will correctly induce high-return behavior. Empirically, we show that DoC is able to achieve significantly better performance than DT on environments that have highly stochastic rewards and transitions.[1]

## 1 Introduction

Offline reinforcement learning (RL) aims to extract an optimal policy solely from an existing dataset of previous interactions (Fujimoto et al., 2019; Wu et al., 2019; Kumar et al., 2020). As researchers begin to scale offline RL to large image, text, and video datasets (Agarwal et al., 2020; Fan et al., 2022; Baker et al., 2022; Reed et al., 2022; Reid et al., 2022), a family of methods known as *return-conditioned supervised learning* (RCSL), including Decision Transformer (DT) (Chen et al., 2021; Lee et al., 2022) and RL via Supervised Learning (RvS) (Emmons et al., 2021), have gained popularity due to their algorithmic simplicity and ease of scaling. At the heart of RCSL is the idea of conditioning a policy on a specific future outcome, often a return (Srivastava et al., 2019; Kumar et al., 2019; Chen et al., 2021) but also sometimes a goal state or generic future event (Codevilla et al., 2018; Ghosh et al., 2019; Lynch et al., 2020). RCSL trains a policy to imitate actions associated with a conditioning input via supervised learning. During inference (i.e., at evaluation), the policy is conditioned on a desirable high-return or future outcome, with the hope of inducing behavior that can achieve this desirable outcome.

---

[1] Code available at `https://github.com/google-research/google-research/tree/master/dichotomy_of_control`.

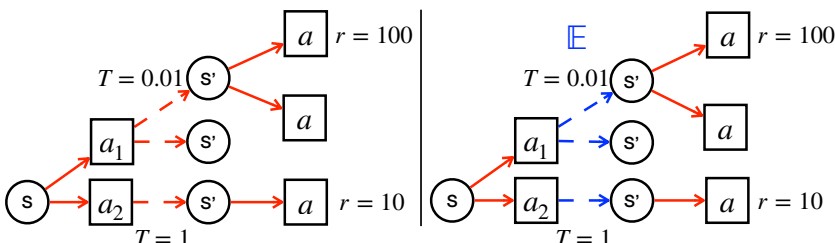

Figure 1: Illustration of DT (RCSL) and DoC. Circles and squares denote states and actions. Solid arrows denote policy decisions. Dotted arrows denote (stochastic) environment transitions. All arrows and nodes are present in the dataset, i.e., there are 4 trajectories, 2 of which achieve 0 reward. DT maximizes returns across an entire trajectory, leading to suboptimal policies when a large return ($r = 100$) is achieved only due to very low-probability environment transitions ($T = 0.01$). DoC separates policy stochasticity from that of the environment and only tries to control action decisions (solid arrows), achieving optimal control through maximizing expected returns at each timestep.

Despite the empirical advantages that come with supervised training (Emmons et al., 2021; Kumar et al., 2021), RCSL can be highly suboptimal in stochastic environments (Paster et al., 2022; Brand-fonbrener et al., 2022), where the future an RCSL policy conditions on (e.g., return) can be primarily determined by randomness in the environment rather than the data collecting policy itself. Figure 1 (left) illustrates an example, where conditioning an RCSL policy on the highest return observed in the dataset ($r = 100$) leads to a policy ($a_1$) that relies on a stochastic transition of very low probability ($T = 0.01$) to achieve the desired return of $r = 100$; by comparison the choice of $a_2$ is much better in terms of average return, as it surely achieves $r = 10$. The crux of the issue is that the RCSL policy is *inconsistent* with its conditioning input. Conditioning the policy on a desired return (i.e., 100) to act in the environment leads to a distribution of real returns (i.e., $0.01 * 100$) that is wildly different from the return value being conditioned on. This issue would not have occurred if the policy could also maximize the transition probability that led to the high-return state, but this is not possible as transition probabilities are a part of the environment and not subject to the policy's control.

A number of works propose a generalization of RCSL, known as future-conditioned supervised learning methods. These techniques have been shown to be effective in imitation learning (Singh et al., 2020; Pertsch et al., 2020), offline Q-learning (Ajay et al., 2020), and online policy gradi-ent (Venuto et al., 2021). It is common in future-conditioned supervised learning to apply a KL divergence regularizer on the latent variable – inspired by variational auto-encoders (VAE) (Kingma & Welling, 2013) and measured with respect to a learned prior conditioned only on past informa-tion – to limit the amount of future information captured in the latent variable. It is natural to ask whether this regularizer could remedy the insconsistency of RCSL. Unfortunately, as the KL regu-larizer makes no distinction between future information that is controllable versus that which is not, such an approach will still exhibit inconsistency, in the sense that the latent variable representation may contain information about the future that is due only to environment stochasticity.

It is clear that the major issue with both RCSL and naïve variational methods is that they make no distinction between stochasticity of the policy (controllable) and stochasticity of the environment (uncontrollable) (Paster et al., 2020; Štrupl et al., 2022). An optimal policy should maximize over the controllable (actions) and take expectations over uncontrollable (e.g., transitions) as shown in Figure 1 (right). This implies that, under a variational approach, the latent variable representation that a policy conditions on should not incorporate any information that is solely due to randomness in the environment. In other words, while the latent representation can and should include informa-tion about future behavior (i.e., actions), it should not reveal any information about the rewards or transitions associated with this behavior.

To this end, we propose a future-conditioned supervised learning framework termed *dichotomy of control* (DoC), which, in Stoic terms (Shapiro, 2014), has *"the serenity to accept the things it cannot*

*change, courage to change the things it can, and wisdom to know the difference."* DoC separates mechanisms within a policy's control (actions) from those beyond a policy's control (environment stochasticity). To achieve this separation, we condition the policy on a latent variable representation of the future while minimizing the mutual information between the latent variable and future stochastic rewards and transitions in the environment. DoC only captures information from the controllable actions and avoids capturing information from the uncontrollable environment transitions in the latent variable so that maximization only happens with respect to the controllable actions. Theoretically, we show that DoC policies are *consistent* with their conditioning inputs, ensuring that conditioning on a high-return future will correctly induce high-return behavior. Empirically, we show that DoC can outperform both RCSL and naïve variational methods on highly stochastic environments.

## 2 RELATED WORK

**Return-Conditioned Supervised Learning.** Since offline RL algorithms (Fujimoto et al., 2019; Wu et al., 2019; Kumar et al., 2020) can be sensitive to hyper-parameters and difficult to apply in practice (Emmons et al., 2021; Kumar et al., 2021), return-conditioned supervised learning (RCSL) has become a popular alternative, particularly when the environment is deterministic and near-expert demonstrations are available (Brandfonbrener et al., 2022). RCSL learns to predict behaviors (actions) by conditioning on desired returns (Schmidhuber, 2019; Kumar et al., 2019) using an MLP policy (Emmons et al., 2021) or a transformer-based policy that encapsulates history (Chen et al., 2021). Richer information other than returns, such as goals (Codevilla et al., 2018; Ghosh et al., 2019) or trajectory-level aggregates (Furuta et al., 2021), have also been used as inputs to a conditional policy in practice. Our work also conditions policies on richer trajectory-level information in the form of a latent variable representation of the future, with additional theoretical justifications of such conditioning in stochastic environments.

**RCSL Failures in Stochastic Environments.** Despite the empirical success of RCSL achieved by DT and RvS, recent work has noted the failure modes in stochastic environments. Paster et al. (2020) and Štrupl et al. (2022) presented counter-examples where online RvS can diverge in stochastic environments. Paster et al. (2022) first identified the failure of return-conditioned supervised learning with stochastic transitions and proposed to cluster offline trajectories and condition the policy on the average cluster returns. While conditioning on expected as opposed to maximum returns is more reasonable, this approach has technical limitations (see Appendix C), and can lead to undesirable policy-averaging, i.e., a single policy covering two very different behaviors (clusters) that happen to have the same return. Brandfonbrener et al. (2022) also identified near-determinism as a necessary condition for RCSL to achieve optimality guarantees similar to other offline RL algorithms but did not propose a solution for RCSL in stochastic settings. Villaflor et al. (2022) also identifies overly optimistic behavior of DT and proposes to use discrete $\beta$-VAE to induce diverse future predictions a policy can condition on. This approach only differs the issue with stochastic environments to stochastic latent variables, i.e., the latent variables will still contain stochastic environment information that the policy cannot reliably reproduce.

**Learning Latent Variables from Offline Data.** Various works have explored learning a latent variable representation of the future (or past) transitions in offline data via maximum likelihood and use the latent variable to assist planning (Lynch et al., 2020), imitation learning (Kipf et al., 2019; Ajay et al., 2020; Hakhamaneshi et al., 2021), offline RL (Ajay et al., 2020; Zhou et al., 2020), or online RL (Fox et al., 2017; Krishnan et al., 2017; Goyal et al., 2019; Shankar & Gupta, 2020; Singh et al., 2020; Wang et al., 2021; Venuto et al., 2021). These works generally focus on the benefit of increased temporal abstraction afforded by using latent variables as higher-level actions in a hierarchical policy. Villaflor et al. (2022) has introduced latent variable models into RCSL, which is one of the essential tools that enables our method, but they did not incoporate the appropriate constraints which can allow RCSL to effectively combat environment stochasticity, as we will see in our work. In general, existing work in latent variable models for future-conditioned supervised learning provides no theoretical guarantees in the literature, whereas our approach provides consistency guarantees. Lastly, Dietterich et al. (2018) used mutual information constraints to separate controllable from uncontrollable aspects of an MDP with the goal of accelerating reinforcement learning, whereas we study the dichotomy of control under the context of return-and-future conditioned supervised learning.

## 3 PRELIMINARIES

**Environment Notation**   We consider the problem of learning a decision-making agent to interact with a sequential, finite-horizon environment. At time $t = 0$, the agent observes an initial state $s_0$ determined by the environment. After observing $s_t$ at a timestep $0 \leq t \leq H$, the agent chooses an action $a_t$. After the action is applied the environment yields an immediate scalar reward $r_t$ and, if $t < H$, a next state $s_{t+1}$. We use $\tau := (s_t, a_t, r_t)_{t=0}^H$ to denote a generic *episode* generated from interactions with the environment, and use $\tau_{i:j} := (s_t, a_t, r_t)_{t=i}^j$ to denote a generic *sub-episode*, with the understanding that $\tau_{0:-1}$ refers to an empty sub-episode. The *return* associated with an episode $\tau$ is defined as $R(\tau) := \sum_{t=0}^H r_t$.

We will use $\mathcal{M}$ to denote the environment. We assume that $\mathcal{M}$ is determined by a stochastic reward function $\mathcal{R}$, stochastic transition function $\mathcal{T}$, and unique initial state $s_0$, so that $r_t \sim \mathcal{R}(\tau_{0:t-1}, s_t, a_t)$ and $s_{t+1} \sim \mathcal{T}(\tau_{0:t-1}, s_t, a_t)$ during interactions with the environment. Note that these definitions specify a history-dependent environment, as opposed to a less general Markovian environment.

**Learning a Policy in RCSL**   In future- or return-conditioned supervised learning, one uses a fixed training data distribution $\mathcal{D}$ of episodes $\tau$ (collected by unknown and potentially multiple agents) to learn a policy $\pi$, where $\pi$ is trained to predict $a_t$ conditioned on the history $\tau_{0:t-1}$, the observation $s_t$, and an additional conditioning variable $z$ that may depend on both the past and future of the episode. For example, in return-conditioned supervised learning, policy training minimizes the following objective over $\pi$:

$$\mathcal{L}_{\text{RCSL}}(\pi) := \mathbb{E}_{\tau \sim \mathcal{D}} \left[ \sum_{t=0}^H - \log \pi(a_t | \tau_{0:t-1}, s_t, z(\tau)) \right], \tag{1}$$

where $z(\tau)$ is the return $R(\tau)$.

**Inconsistency of RCSL**   To apply an RCSL-trained policy $\pi$ during inference — i.e., interacting online with the environment — one must first choose a specific $z$.[2] For example, one might set $z$ to be the maximal return observed in the dataset, in the hopes of inducing a behavior policy which achieves this high return. Using $\pi_z$ as a shorthand to denote the policy $\pi$ conditioned on a specific $z$, we define the expected return $V_{\mathcal{M}}(\pi_z)$ of $\pi_z$ in $\mathcal{M}$ as,

$$V_{\mathcal{M}}(\pi_z) := \mathbb{E}_{\tau \sim \text{Pr}[\cdot | \pi_z, \mathcal{M}]} [R(\tau)]. \tag{2}$$

Ideally the expected return induced by $\pi_z$ is close to $z$, i.e., $z \approx V_{\mathcal{M}}(\pi_z)$, so that acting according to $\pi$ conditioned on a high return induces behavior which actually achieves a high return. However, RCSL training according to Equation 1 will generally yield policies that are highly *inconsistent* in stochastic environments, meaning that the achieved returns may be significantly different than $z$ (i.e., $V_{\mathcal{M}}(\pi_z) \neq z$). This has been highlighted in various previous works (Brandfonbrener et al., 2022; Paster et al., 2022; Štrupl et al., 2022; Eysenbach et al., 2022; Villaflor et al., 2022), and we provided our own example in Figure 1.

**Approaches to Mitigating Inconsistency**   A number of future-conditioned supervised learning approaches propose to learn a stochastic latent variable embedding of the future, $q(z|\tau)$, while regularizing $q$ with a KL-divergence from a learnable *prior* conditioned only on the past $p(z|s_0)$ (Ajay et al., 2020; Venuto et al., 2021; Lynch et al., 2020), thereby minimizing:

$$\mathcal{L}_{\text{VAE}}(\pi, q, p) := \mathbb{E}_{\tau \sim \mathcal{D}, z \sim q(z|\tau)} \left[ \sum_{t=0}^H - \log \pi(a_t | \tau_{0:t-1}, s_t, z) \right] + \beta \cdot \mathbb{E}_{\tau \sim \mathcal{D}} [D_{\text{KL}}(q(z|\tau) \| p(z|s_0))]. \tag{3}$$

One could consider adopting such a future-conditioned objective in RCSL. However, since the KL regularizer makes no distinction between observations the agent can control (actions) from those it cannot (environment stochasticity), the choice of coefficient $\beta$ applied to the regularizer introduces

---

[2]For simplicitly, we assume $z$ is chosen at timestep $t = 0$ and held constant throughout an entire episode. As noted in Brandfonbrener et al. (2022), this protocol also encompasses instances like DT (Chen et al., 2021) in which $z$ at timestep $t$ is the (desired) return summed starting at $t$.

a 'lose-lose' trade-off. Namely, as noted in Ajay et al. (2020), if the regularization coefficient is too large ($\beta \geq 1$), the policy will not learn diverse behavior (since the KL limits how much information of the future actions is contained in $z$); while if the coefficient is too small ($\beta < 1$), the policy's learned behavior will be inconsistent (in the sense that $z$ will contain information of environment stochasticity that the policy cannot reliably reproduce). The discrete $\beta$-VAE incoporated by Villaflor et al. (2022) with $\beta < 1$ corresponds to this second failure mode.

## 4 DICHOTOMY OF CONTROL

In this section, we first propose the DoC objective for learning future-conditioned policies that are guaranteed to be consistent. We then present a practical framework for optimizing DoC's constrained objective in practice and an inference scheme to enable better-than-dataset behavior via a learned value function and prior.

### 4.1 DICHOTOMY OF CONTROL VIA MUTUAL INFORMATION MINIMIZATION

As elaborated in the prevous section, whether $z(\tau)$ is the return $R(\tau)$ or more generally a stochastic latent variable with distribution $q(z|\tau)$, existing RCSL methods fail to satisfy consistency because they insufficiently enforce the type of future information $z$ can contain. A key observation is that $z$ should not include any information due to environment stochasticity, i.e., any information about a future $r_t, s_{t+1}$ that is not already known given the previous history up to that point $\tau_{0:t-1}, s_t, a_t$. (A similar observation was made in (Paster et al., 2022) under the more restrictive assumption that $z$ is a cluster index, which we do not require here.) To address the independence requirement in a general and sound way, we modify the RCSL objective from Equation 1 to incorporate a conditional mutual information constraint between $z$ and each pair $r_t, s_{t+1}$ in the future:

$$\mathcal{L}_{\text{DoC}}(\pi, q) := \mathbb{E}_{\tau \sim \mathcal{D}, z \sim q(z|\tau)} \left[ \sum_{t=0}^{H} -\log \pi(a_t | \tau_{0:t-1}, s_t, z) \right]$$

$$\text{s.t.} \quad \text{MI}(r_t; z \mid \tau_{0:t-1}, s_t, a_t) = 0, \text{MI}(s_{t+1}; z \mid \tau_{0:t-1}, s_t, a_t) = 0, \quad (4)$$

$$\forall \tau_{0:t-1}, s_t, a_t \text{ and } 0 \leq t \leq H, \quad (5)$$

where $\text{MI}(r_t; z|\tau_{0:t-1}, s_t, a_t)$ denotes the mutual information between $r_t$ and $z$ given $\tau_{0:t-1}, s_t, a_t$ when measured under samples of $r_t, z$ from $\mathcal{D}, q$; and analogously for $\text{MI}(s_{t+1}; z|\tau_{0:t-1}, s_t, a_t)$.

The first part of the DoC objective conditions the policy on a latent variable representation of the future, similar to the first part of the future-conditioned VAE objective in Equation 3. However, unlike Equation 3, the DoC objective enforces a much more precise constraint on $q$, given by the MI constraints in Equation 4.

### 4.2 DICHOTOMY OF CONTROL IN PRACTICE

**Contrastive Learning of DoC Constraints.** To satisfy the mutual information constraints in Equation 4 we transform the MI to a contrastive learning objective. Specifically, for the constraint on $r$ and $z$ (and similarly on $s_{t+1}$ and $z$) one can derive,

$$\text{MI}(r_t; z|\tau_{0:t-1}, s_t, a_t)$$
$$= D_{\text{KL}} \left( \Pr[r_t, z|\tau_{0:t-1}, s_t, a_t] \| \Pr[r_t|\tau_{0:t-1}, s_t, a_t] \Pr[z|\tau_{0:t-1}, s_t, a_t] \right)$$
$$= \mathbb{E}_{\Pr[r_t, z|\tau_{0:t-1}, s_t, a_t]} \left[ \log \left( \frac{\Pr[r_t|z, \tau_{0:t-1}, s_t, a_t]}{\Pr[r_t|\tau_{0:t-1}, s_t, a_t]} \right) \right]$$
$$= \mathbb{E}_{\Pr[r_t, z|\tau_{0:t-1}, s_t, a_t]} \log \Pr[r_t|z, \tau_{0:t-1}, s_t, a_t] - \mathbb{E}_{\Pr[r_t|\tau_{0:t-1}, s_t, a_t]} \log \Pr[r_t|\tau_{0:t-1}, s_t, a_t]. \quad (6)$$

The second expectation above is a constant with respect to $z$ and so can be ignored during learning. We further introduce a conditional distribution $\omega(r_t|\tau_{0:t-1}, s_t, a_t)$ parametrized by an energy-based function $f : \Omega \mapsto \mathbb{R}$:

$$\omega(r_t|z, \tau_{0:t-1}, s_t, a_t) \propto \rho(r_t) \exp \left\{ f(r_t, z, \tau_{0:t-1}, s_t, a_t) \right\}, \quad (7)$$

where $\rho$ is some fixed sampling distribution of rewards. In practice, we set $\rho$ to be the marginal distribution of rewards in the dataset. Hence we express the first term of Equation 6 via an optimization

over $\omega$, i.e.,

$$\max_{\omega} \mathbb{E}_{\Pr[r_t, z | \tau_{0:t-1}, s_t, a_t]} \left[ \log \omega(r_t | \tau_{0:t-1}, s_t, a_t) \right]$$

$$= \max_{f} \mathbb{E}_{\Pr[r_t, z | \tau_{0:t-1}, s_t, a_t]} \left[ f(r_t, z, \tau_{0:t-1}, s_t, a_t) - \log \mathbb{E}_{\rho(\tilde{r})} \left[ \exp\{ f(\tilde{r}, z, \tau_{0:t-1}, s_t, a_t) \} \right] \right].$$

Combining this (together with the analogous derivation for $\text{MI}(s_{t+1}; z | \tau_{0:t-1}, s_t, a_t)$) with Equation 4 via the Lagrangian, we can learn $\pi$ and $q(z|\tau)$ by minimizing the final DoC objective:

$$\mathcal{L}_{\text{DoC}}(\pi, q) = \max_{f} \mathbb{E}_{\tau \sim \mathcal{D}, z \sim q(z|\tau)} \left[ \sum_{t=0}^{H} - \log \pi(a_t | \tau_{0:t-1}, s_t, z) \right]$$

$$+ \beta \cdot \sum_{t=0}^{H} \mathbb{E}_{\tau \sim \mathcal{D}, z \sim q(z|\tau)} \left[ f(r_t, s_{t+1}, z, \tau_{0:t-1}, s_t, a_t) - \log \mathbb{E}_{\rho(\tilde{r}, \tilde{s}')} \left[ \exp\{ f(\tilde{r}, \tilde{s}', z, \tau_{0:t-1}, s_t, a_t) \} \right] \right].$$

(8)

---

**Algorithm 1** Inference with Dichotomy of Control

---

**Inputs** Policy $\pi(\cdot | \cdot, \cdot, \cdot)$, prior $p(\cdot)$, value function $V(\cdot)$, initial state $s_0$, number of samples hyperparameter $K$.
Initialize $z^*; V^*$                 ▷ Track the best latent and its value.
**for** $k = 1$ to $K$ **do**
    Sample $z_k \sim p(z|s_0)$         ▷ Sample a latent from the learned prior.
    **if** $V(z_k) > V^*$ **then**
        $z^* = z_k; V^* = V$    ▷ Set best latent to the one with the highest value.
**return** $\pi(\cdot | \cdot, \cdot, z^*)$             ▷ Policy conditioned on the best $z^*$.

---

**DoC Inference.** As is standard in RCSL approaches, the policy learned by DoC requires an appropriate conditioning input $z$ to be chosen during inference. To choose a desirable $z$ associated with high return, we propose to (1) enumerate or sample a large number of potential values of $z$, (2) estimate the expected return for each of these values of $z$, (3) choose the $z$ with the highest associated expected return to feed into the policy. To enable such an inference-time procedure, we need to add two more components to the method formulation: First, a prior distribution $p(z|s_0)$ from which we will sample a large number of values of $z$; second, a value function $V(z)$ with which we will rank the potential values of $z$. These components are learned by minimizing the following objective:

$$\mathcal{L}_{\text{aux}}(V, p) = \mathbb{E}_{\tau \sim \mathcal{D}, z \sim q(z|\tau)} \left[ (V(z) - R(\tau))^2 + D_{\text{KL}}(\text{stopgrad}(q(z|\tau)) \| p(z|s_0)) \right]. \quad (9)$$

Note that we apply a stop-gradient to $q(z|\tau)$ when learning $p$ so as to avoid regularizing $q$ with the prior. This is unlike the VAE approach, which by contrast advocates *for* regularizing $q$ via the prior. See Algorithm 1 for inference pseudocode (and Appendix D for training pseudocode). Since DoC requires sampling multiple latents, it incurs additional computational overhead compared to vanilla DT. We differ improving the computational efficiency of DoC to future work.

## 5 Consistency Guarantees for Dichotomy of Control

We provide a theoretical justification of the proposed learning objectives $\mathcal{L}_{\text{DoC}}$ and $\mathcal{L}_{\text{aux}}$, showing that, if they are minimized, the resulting inference-time procedure will be sound, in the sense that DoC will learn a $V$ and $\pi$ such that the true value of $\pi_z$ in the environment $\mathcal{M}$ is equal to $V(z)$. More specifically we define the following notion of *consistency*:

**Definition 1** (Consistency). *A future-conditioned policy $\pi$ and value function $V$ are* **consistent** *for a specific conditioning input $z$ if the expected return of $z$ predicted by $V$ is equal to the true expected return of $\pi_z$ in the environment: $V(z) = V_{\mathcal{M}}(\pi_z)$.*

To guarantee consistency of $\pi, V$, we will make the following two assumptions:

**Assumption 2** (Data and environment agreement). *The per-step reward and next-state transitions observed in the data distribution are the same as those of the environment. In other words,*

*for any* $\tau_{0:t-1}, s_t, a_t$ *with* $\Pr[\tau_{0:t-1}, s_t, a_t | \mathcal{D}] > 0$, *we have* $\Pr[\hat{r}_t = r_t | \tau_{0:t-1}, s_t, a_t, \mathcal{D}] = \mathcal{R}(\hat{r}_t | \tau_{0:t-1}, s_t, a_t)$ *and* $\Pr[\hat{s}_{t+1} = s_{t+1} | \tau_{0:t-1}, s_t, a_t, \mathcal{D}] = \mathcal{T}(\hat{s}_{t+1} | \tau_{0:t-1}, s_t, a_t)$ *for all* $\hat{r}_t, \hat{s}_{t+1}$.

**Assumption 3** (No optimization or approximation errors). *DoC yields policy* $\pi$ *and value function* $V$ *that are Bayes-optimal with respect to the training data distribution and* $q$. *In other words,* $V(z) = \mathbb{E}_{\tau \sim \Pr[\cdot | z, \mathcal{D}]} [R(\tau)]$ *and* $\pi(\hat{a} | \tau_{0:t-1}, s_t, z) = \Pr[\hat{a} = a_t | \tau_{0:t-1}, s_t, z, \mathcal{D}]$.

Given these two assumptions, we can then establish the following consistency guarantee for DoC.

**Theorem 4.** *Suppose DoC yields* $\pi, V, q$ *with* $q$ *satisfying the MI constraints:*

$$\mathrm{MI}(r_t; z | \tau_{0:t-1}, s_t, a_t) = \mathrm{MI}(s_{t+1}; z | \tau_{0:t-1}, s_t, a_t) = 0, \tag{10}$$

*for all* $\tau_{0:t-1}, s_t, a_t$ *with* $\Pr[\tau_{0:t-1}, s_t, a_t | \mathcal{D}] > 0$. *Then under Assumptions 2 and 3,* $V$ *and* $\pi$ *are consistent for any* $z$ *with* $\Pr[z | q, \mathcal{D}] > 0$.

For proof, see Appendix A.

**Consistency in Markovian environments.** While the results above are focused on environments and policies that are non-Markovian, one can extend Theorem 4 to Markovian environments and policies. This result is somewhat surprising, as the assignments of $z$ to episodes $\tau$ induced by $q$ are necessarily history-dependent, and projecting the actions appearing in these clusters to a non-history-dependent policy would seemingly lose important information. However, a Markovian assumption on the rewards and transitions of the environment is sufficient to ensure that no 'important' information will be lost, at least in terms of the satisfying requirements for consistency in Definition 1. Alternative notions of consistency are not as generally applicable; see Appendix C.

We begin by stating our assumptions.

**Assumption 5** (Markov environment). *The rewards and transitions of* $\mathcal{M}$ *are Markovian; i.e.,* $\mathcal{R}(\tau_{0:t-1}, s_t, a_t) = \mathcal{R}(\tilde{\tau}_{0:t-1}, s_t, a_t)$ *and* $\mathcal{T}(\tau_{0:t-1}, s_t, a_t) = \mathcal{T}(\tilde{\tau}_{0:t-1}, s_t, a_t)$ *for all* $\tau, \tilde{\tau}, s_t, a_t$. *We use the shorthand* $\mathcal{R}(s_t, a_t), \mathcal{T}(s_t, a_t)$ *for these history-independent functions.*

**Assumption 6** (Markov policy, without optimization or approximation errors). *The policy learned by DoC is Markov. This policy* $\pi$ *as well as its corresponding learned value function* $V$ *are Bayes-optimal with respect to the training data distribution and* $q$. *In other words,* $V(z) = \mathbb{E}_{\tau \sim \Pr[\cdot | z, \mathcal{D}]} [R(\tau)]$ *and* $\pi(\hat{a} | s_t, z) = \Pr[\hat{a} = a_t | s_t, z, \mathcal{D}]$.

With these two assumptions, we can then establish the analogue to Theorem 4, which relaxes the dependency on history for both the policy $\pi$ and the MI constraints:

**Theorem 7.** *Suppose DoC yields* $\pi, V, q$ *with* $q$ *satisfying the MI constraints:*

$$\mathrm{MI}(r_t; z | s_t, a_t) = \mathrm{MI}(s_{t+1}; z | s_t, a_t) = 0, \tag{11}$$

*for all* $s_t, a_t$ *with* $\Pr[s_t, a_t | \mathcal{D}] > 0$. *Then under Assumptions 2, 5, and 6,* $V$ *and* $\pi$ *are consistent for any* $z$ *with* $\Pr[z | q, \mathcal{D}] > 0$.

For proof, see Appendix B.

## 6 EXPERIMENTS

We conducted an empirical evaluation to ascertain the effectiveness of DoC. For this evaluation, we considered three settings: (1) a Bernoulli bandit problem with stochastic rewards, based on a canonical 'worst-case scenario' for RCSL (Brandfonbrener et al., 2022); (2) the FrozenLake domain from (Brockman et al., 2016), where the future VAE approach proves ineffective; and finally (3) a modified set of OpenAI Gym (Brockman et al., 2016) environments where we introduced environment stochasticity. In these studies, we found that DoC exhibits a significant advantage over RCSL/DT, and outperforms future VAE when the analogous to "one-step" RL is insufficient. We consider vanilla DT (Chen et al., 2021), which trains a "returns-to-go, state, action" transformer. For DoC and the VAE baseline, we adopt similar inference scheme as (Lee et al., 2022), where multiple latent is sampled to determine the best next action. For DT, we use the same implementation and hyperparameters as Chen et al. (2021). Both VAE and DoC are built upon the DT implementation and additionally learn a Gaussian latent variable over succeeding 20 future steps. See experiment details in Appendix E and additional results in Appendix F.

### 6.1 EVALUATING STOCHASTIC REWARDS IN BERNOULLI BANDIT

**Bernoulli Bandit.** Consider a two-armed bandit as shown in Figure 2 (left). The two arms, $a_1, a_2$, have stochastic rewards drawn from Bernoulli distributions of $\text{Bern}(1-p)$ and $\text{Bern}(p)$, respectively. In the offline dataset, the $a_1$ arm with reward $\text{Bern}(1-p)$ is pulled with probability $\pi_D(a_1) = p$. When $p$ is small, this corresponds to the better arm only being pulled occasionally. Under this setup, $\pi_{\text{RCSL}}(a_1|r=1) = \pi_{\text{RCSL}}(a_2|r=1) = 0.5$, which is highly suboptimal compared to always pulling the optimal arm $a_1$ with reward $\text{Bern}(1-p)$ for $p < 0.5$.

**Results.** We train tabular DoC and baselines on 1000 samples where the superior arm with $r \sim \text{Bern}(1-p)$ is pulled with probability $p$ for $p \in \{0.1, ..., 0.5\}$. Figure 2 (right) shows that RCSL and percentage BC (filtered by $r = 1$) always result in policies that are indifferent in the arms, whereas DoC is able to recover the Bayes-optimal performance (dotted line) for all $p$ values considered. Future VAE performs similarly to DoC for small $p$ values, but is sensitive to the KL regularization coefficient when $p$ is close to $0.5$.

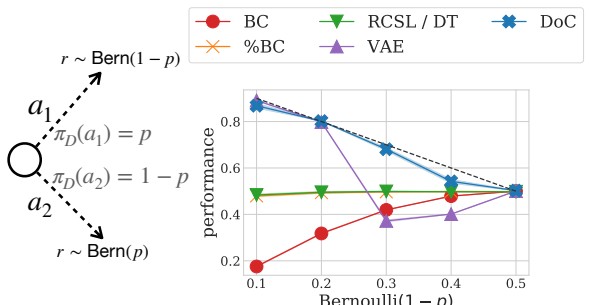

Figure 2: [Left] Bernoulli bandit where the better arm $a_1$ with reward $\text{Bern}(1-p)$ for $p < 0.5$ is pulled with probability $\pi_D(a_1) = p$ in the offline data. [Right] Average rewards achieved by DoC and baselines across 5 environment seeds. RCSL is highly suboptimal when $p$ is small, whereas DoC achieves close to Bayes-optimal performance (dotted line) for all values of $p$.

### 6.2 EVALUATING STOCHASTIC TRANSITIONS IN FROZENLAKE

**FrozenLake.** Next, we consider the FrozenLake environment with stochastic transitions where the agent taking an action has probability $p$ of moving in the intended direction, and probability $0.5 \cdot (1-p)$ of slipping to either of the two sides of the intended direction. We collect 100 trajectories of length 100 using a DQN policy trained in the original environment ($p = \frac{1}{3}$) which achieves an average return of $0.7$, and vary $p$ during data collection and evaluation to test different levels of stochasticity. We also include uniform actions with probability $\epsilon$ to lower the performance of the offline data so that BC is highly suboptimal.

**Results.** Figure 3 presents the visualization (left) and results (right) for this task. When the offline data is closer to being expert ($\epsilon = 0.3$), DT, future VAE, and DoC perform similarly with better performance in more deterministic environments. As the offline dataset becomes more suboptimal ($\epsilon = 0.5$), DoC starts to dominate across all levels of transition stochasticity. When the offline data is highly suboptimal ($\epsilon = 0.7$), DT and future VAE has little advantage over BC, whereas DoC continues to learn policies with reasonable performance.

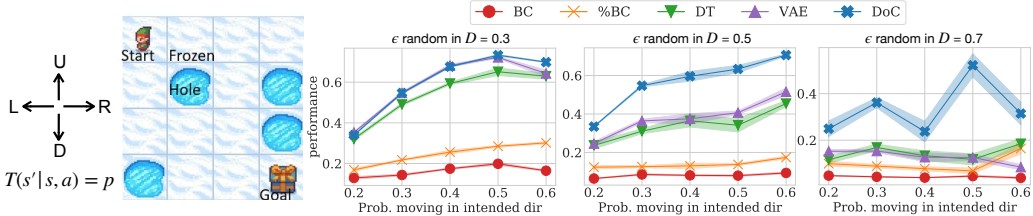

Figure 3: [Left] Visualization of the stochastic FrozenLake task. The agent has a probability $p$ of moving in the intended direction and $1-p$ of slipping to either sides. [Right] Average performance (across 5 seeds) of DoC and baselines on FrozenLake with different levels of stochasticity ($p$) and offline dataset quality ($\epsilon$). DoC outperforms DT and future VAE, where the gain is more salient when the offline data is less optimal ($\epsilon = 0.5$ and $\epsilon = 0.7$).

Figure 4: Average performance (across 5 seeds) of DoC and baselines on modified stochastic Gym MuJoCo and AntMaze tasks. DoC and future VAE generally provide benefits over DT, where DoC provide more benefits on harder tasks such as Humanoid. Future VAE can be sensitive to the KL coefficient $\beta$, which can result in the failure mode shown in Reacher-v2 if not tuned properly.

## 6.3 EVALUATING STOCHASTIC GYM MUJOCO

**Environments.** We now consider a set of Gym MuJoCo environments including Reacher, Hopper, HalfCheetah, and Humanoid. We additionally consider AntMaze from D4RL (Fu et al., 2020). These environments are deterministic by default, which we modify by introducing time-correlated Gaussian noise to the actions before inputing the action into the physics simulator during data collection and evaluation for all but AntMaze environments. Specifically, the Gaussian noise we introduce to the actions has 0 mean and standard deviation of the form $(1 - e^{-0.01 \cdot t}) \cdot \sin(t) \cdot \sigma$ where $t$ is the step number and $\sigma \in [0, 1]$. For AntMaze where the dataset has already been collected in the deterministic environment by D4RL, we add gaussian noise with 0.1 standard deviation to the reward uniformly with probability 0.1 (both to the dataset and during evaluation).

**Results.** Figure 4 shows the average performance (across 5 seeds) of DT, future VAE, and DoC on these stochastic environments. Both future VAE and DoC generally provide benefits over DT, where the benefit of DoC is more salient in harder environments such as HalfCheetah and Humanoid. We found future VAE to be sensitive to the $\beta$ hyperparameter, and simply using $\beta = 1$ can result in the falure case as shown in Reacher-v2.

## 7 CONCLUSION

Despite the empirical promise of return- or future-conditioned supervised learning (RCSL) with large transformer architectures, environment stochasticity hampers the application of supervised learning to sequential decision making. To address this issue, we proposed to augment supervised learning with the dichotomy of control principle (DoC), guiding a supervised policy to only control the controllable (actions). Theoretically, DoC learns *consistent* policies, guaranteeing that they achieve the future or return they are conditioned on. Empirically, DoC outperforms RCSL in highly stochastic environments. While DoC still falls short in addressing other RL challenges such as 'stitching' (i.e., composing sub-optimal trajectories), we hope that dichotomy of control serves as a stepping stone in solving sequential decision making with large-scale supervised learning.

### ACKNOWLEDGMENTS

Thanks to George Tucker for reviewing draft versions of this manuscript. Thanks to Marc Bellemare for help with derivations. Thanks to David Brandfonbrener and Keiran Paster for discussions around stochastic environments. We gratefully acknowledges the support of a Canada CIFAR AI Chair, NSERC and Amii, and support from Berkeley BAIR industrial consortion.

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

# Appendix

## A    PROOF OF THEOREM 4

The proof relies on the following lemma, showing that the MI constraints ensure that the observed rewards and dynamics conditioned on $z$ in the training data are equal to the rewards and dynamics of the environment.

**Lemma 8.** *Suppose DoC yields $q$ satisfying the MI constraints:*

$$\mathrm{MI}(r_t; z|\tau_{0:t-1}, s_t, a_t) = \mathrm{MI}(s_{t+1}; z|\tau_{0:t-1}, s_t, a_t) = 0, \tag{12}$$

*for all $\tau_{0:t-1}, s_t, a_t$ with $\Pr[\tau_{0:t-1}, s_t, a_t|\mathcal{D}] > 0$. Then under Assumption 2,*

$$\Pr\left[\hat{r} = r_t \mid \tau_{0:t-1}, s_t, a_t, z, \mathcal{D}\right] = \mathcal{R}(\hat{r}_t \mid \tau_{0:t-1}, s_t, a_t), \tag{13}$$
$$\Pr\left[\hat{s}_{t+1} = s_{t+1} \mid \tau_{0:t-1}, s_t, a_t, z, \mathcal{D}\right] = \mathcal{T}(\hat{s}_{t+1} \mid \tau_{0:t-1}, s_t, a_t), \tag{14}$$

*for all $\tau_{0:t-1}, s_t, a_t, z$ and $\hat{r}, \hat{s}_{t+1}$, as long as $\Pr[\tau_{0:t-1}, s_t, a_t, z|\mathcal{D}] > 0$.*

*Proof.* We show the derivations relevant to reward, with those for next-state being analogous. We start with the definition of mutual information:

$$\mathrm{MI}(r_t; z|\tau_{0:t-1}, s_t, a_t) = \mathbb{E}_{(r_t,z) \sim \Pr[\cdot|\tau_{0:t-1}, s_t, a_t, \mathcal{D}]} \left[ \log \frac{\Pr\left[r_t|\tau_{0:t-1}, s_t, a_t, z, \mathcal{D}\right]}{\Pr\left[r_t|\tau_{0:t-1}, s_t, a_t, \mathcal{D}\right]} \right] \tag{15}$$

$$= \mathbb{E}_{z \sim \Pr[\cdot|\tau_{0:t-1}, s_t, a_t, \mathcal{D}]} \left[ D_{\mathrm{KL}}(\Pr[r|\tau_{0:t-1}, s_t, a_t, z, \mathcal{D}] \| \Pr[r|\tau_{0:t-1}, s_t, a_t, \mathcal{D}]) \right]. \tag{16}$$

The KL divergence is a nonnegative quantity, and it is zero only when the two input distributions are equal. Thus, the constraint $\mathrm{MI}(r_t; z|\tau_{0:t-1}, s_t, a_t) = 0$ implies,

$$\Pr\left[r|\tau_{0:t-1}, s_t, a_t, z, \mathcal{D}\right] = \Pr[r|\tau_{0:t-1}, s_t, a_t, \mathcal{D}], \tag{17}$$

for all $\tau_{0:t-1}, s_t, a_t, z$ with $\Pr[z|\tau_{0:t-1}, s_t, a_t, \mathcal{D}] > 0$. From Assumption 2 we know

$$\Pr[r|\tau_{0:t-1}, s_t, a_t, \mathcal{D}] = \mathcal{R}(r|\tau_{0:t-1}, s_t, a_t), \tag{18}$$

and so we immediately have the desired result.

We will further employ the following lemma, which takes us most of the way to proving Theorem 4:

**Lemma 9.** *Suppose DoC yields $\pi, q$ with $q$ satisfying the MI constraints:*

$$\mathrm{MI}(r_t; z|\tau_{0:t-1}, s_t, a_t) = \mathrm{MI}(s_{t+1}; z|\tau_{0:t-1}, s_t, a_t) = 0, \tag{19}$$

*for all $\tau_{0:t-1}, s_t, a_t$ with $\Pr[\tau_{0:t-1}, s_t, a_t|\mathcal{D}] > 0$. Then under Assumptions 2 and 3, we have*

$$\Pr\left[\tau \mid z, \mathcal{D}\right] = \Pr\left[\tau \mid \pi_z, \mathcal{M}\right], \tag{20}$$

*for all $\tau$ and all $z$ with $\Pr[z|q, \mathcal{D}] > 0$.*

*Proof.* We may write the probability $\Pr\left[\tau \mid z, \mathcal{D}\right]$ as,

$$\Pr\left[\tau \mid z, \mathcal{D}\right] = \prod_{t=0}^{H} \Pr\left[a_t \mid \tau_{0:t-1}, s_t, z, \mathcal{D}\right]$$

$$\cdot \prod_{t=0}^{H} \Pr\left[r_t \mid \tau_{0:t-1}, s_t, a_t, z, \mathcal{D}\right]$$

$$\cdot \prod_{t=0}^{H-1} \Pr\left[s_{t+1} \mid \tau_{0:t-1}, s_t, a_t, z, \mathcal{D}\right]. \tag{21}$$

**Case 1:** We begin by considering the case of $\tau$ satisfying $\Pr[\tau \mid z, \mathcal{D}] > 0$. For such a $\tau$, by Assumption 3 we may write the first probability above as

$$\Pr[a_t \mid \tau_{0:t-1}, s_t, z, \mathcal{D}] = \pi_z(a_t | \tau_{0:t-1}, s_t). \tag{22}$$

Moreover, by Lemma 8 we may write the second and third probabilities as

$$\Pr[r_t \mid \tau_{0:t-1}, s_t, a_t, z, \mathcal{D}] = \mathcal{R}(r_t | \tau_{0:t-1}, s_t, a_t) \tag{23}$$
$$\Pr[s_{t+1} \mid \tau_{0:t-1}, s_t, a_t, z, \mathcal{D}] = \mathcal{T}(s_{t+1} | \tau_{0:t-1}, s_t, a_t). \tag{24}$$

Therefore, for any $\tau$ with $\Pr[\tau \mid z, \mathcal{D}] > 0$ we have,

$$\Pr[\tau \mid z, \mathcal{D}] = \prod_{t=0}^{H} \pi_z(a_t | \tau_{0:t-1}, s_t) \cdot \prod_{t=0}^{H} \mathcal{R}(r_t | \tau_{0:t-1}, s_t, a_t) \cdot \prod_{t=0}^{H-1} \mathcal{T}(s_{t+1} | \tau_{0:t-1}, s_t, a_t)$$
$$= \Pr[\tau \mid \pi_z, \mathcal{M}]. \tag{25}$$

**Case 2:** To handle the case of $\Pr[\tau \mid z, \mathcal{D}] = 0$ we will show that $\Pr[\tau_{0:t} \mid z, \mathcal{D}] = 0$ implies $\Pr[\tau_{0:t} \mid \pi_z, \mathcal{M}] = 0$ by induction on $t$. The base case of $t = -1$ is trivial. For $t > -1$, we may write,

$$\Pr[\tau_{0:t} \mid z, \mathcal{D}] = \Pr[\tau_{0:t-1} \mid z, \mathcal{D}] \cdot \Pr[s_t \mid \tau_{0:t-2}, s_{t-1}, a_{t-1}, z, \mathcal{D}] \cdot \Pr[a_t \mid \tau_{0:t-1}, s_t, z, \mathcal{D}] \cdot$$
$$\Pr[r_t \mid \tau_{0:t-1}, s_t, a_t, z, \mathcal{D}], \tag{26}$$

$$\Pr[\tau_{0:t} \mid \pi_z, \mathcal{M}] = \Pr[\tau_{0:t-1} \mid \pi_z, \mathcal{M}] \cdot \mathcal{T}(s_t | \tau_{0:t-2}, s_{t-1}, a_{t-1}) \cdot \pi_z(a_t | \tau_{0:t-1}, s_t) \cdot$$
$$\mathcal{R}(r_t | \tau_{0:t-1}, s_t, a_t). \tag{27}$$

Suppose, for the sake of contradiction, that $\Pr[\tau_{0:t} \mid z, \mathcal{D}] = 0$ while $\Pr[\tau_{0:t} \mid \pi_z, \mathcal{M}] > 0$. By the inductive hypothesis, $\Pr[\tau_{0:t-1} \mid z, \mathcal{D}] > 0$. Thus, by Lemma 8 we must have

$$\Pr[s_t \mid \tau_{0:t-2}, s_{t-1}, a_{t-1}, z, \mathcal{D}] = \mathcal{T}(s_t | \tau_{0:t-2}, s_{t-1}, a_{t-1}), \tag{28}$$

and so $\mathcal{T}(s_t | \tau_{0:t-2}, s_{t-1}, a_{t-1}) > 0$ implies that $\Pr[s_t \mid \tau_{0:t-2}, s_{t-1}, a_{t-1}, z, \mathcal{D}] > 0$. Thus, by Assumption 3 we must have

$$\Pr[a_t \mid \tau_{0:t-1}, s_t, z, \mathcal{D}] = \pi_z(a_t | \tau_{0:t-1}, s_t), \tag{29}$$

and so $\pi_z(a_t | \tau_{0:t-1}, s_t) > 0$ implies that $\Pr[a_t \mid \tau_{0:t-1}, s_t, z, \mathcal{D}] > 0$. Lastly, by Lemma 8 we must have
$$\Pr[r_t \mid \tau_{0:t-1}, s_t, a_t, z, \mathcal{D}] = \mathcal{R}(r_t | \tau_{0:t-1}, s_t, a_t), \tag{30}$$

and so $\mathcal{R}(r_t | \tau_{0:t-1}, s_t, a_t) > 0$ implies that $\Pr[r_t \mid \tau_{0:t-1}, s_t, a_t, z, \mathcal{D}] > 0$. Altogether, we find that each of the three terms on the RHS of Equation 26 is strictly positive and so $\Pr[\tau_{0:t} \mid z, \mathcal{D}] > 0$; contradiction.

### A.1 THEOREM PROOF

We are now prepared to prove Theorem 4.

Using Assumption 3, we can express $V(z)$ as,

$$V(z) = \int \Pr[\hat{\tau} = \tau \mid z, \mathcal{D}] \cdot R(\hat{\tau}) \, d\hat{\tau}. \tag{31}$$

By Lemma 9 we have,

$$V(z) = \int \Pr[\hat{\tau} = \tau \mid z, \mathcal{D}] \cdot R(\hat{\tau}) \, d\hat{\tau} \tag{32}$$

$$= \int \Pr[\hat{\tau} = \tau \mid \pi_z, \mathcal{M}] \cdot R(\hat{\tau}) \, d\hat{\tau} \tag{33}$$

$$= V_{\mathcal{M}}(\pi_z), \tag{34}$$

as desired.

# B  PROOF OF THEOREM 7

We begin by proving a result under stricter conditions, namely, when the MI constraints retain the conditioning on history.

**Lemma 10.** *Suppose DoC yields $\pi, V, q$ with $q$ satisfying the MI constraints:*

$$\mathrm{MI}(r_t; z|\tau_{0:t-1}, s_t, a_t) = \mathrm{MI}(s_{t+1}; z|\tau_{0:t-1}, s_t, a_t) = 0, \tag{35}$$

*for all $\tau_{0:t-1}, s_t, a_t$ with $\Pr[\tau_{0:t-1}, s_t, a_t|\mathcal{D}] > 0$. Then under Assumptions 2, 5, and 6, $V$ and $\pi$ are consistent for any $z$ with $\Pr[z|q, \mathcal{D}] > 0$.*

*Proof.* Let

$$\pi_z^{\mathrm{hist}}(\hat{a} \mid \tau_{0:t-1}, s_t) = \Pr\left[\hat{a} = a_t \mid \tau_{0:t-1}, s_t, z, \mathcal{D}\right]. \tag{36}$$

By Lemma 9 and Theorem 4 we have

$$\Pr\left[\tau \mid z, \mathcal{D}\right] = \Pr\left[\tau \mid \pi_z^{\mathrm{hist}}, \mathcal{M}\right], \tag{37}$$

for all $\tau$ and

$$V(z) = V_{\mathcal{M}}(\pi_z^{\mathrm{hist}}), \tag{38}$$

for all $z$ with $\Pr[z \mid q, \mathcal{D}] > 0$.

It is left to show that $V_{\mathcal{M}}(\pi_z^{\mathrm{hist}}) = V_{\mathcal{M}}(\pi_z)$. To do so, we invoke Theorem 5.5.1 in Puterman (2014), which states that, for any history-dependent policy, there exists a Markov policy such that the state-action visitation occupancies of the two policies are equal (and, accordingly, their values are equal). In other words, there exists a Markov policy $\tilde{\pi}_z$ such that

$$\Pr\left[\hat{s} = s_t, \hat{a} = a_t \mid \pi_z^{\mathrm{hist}}, \mathcal{M}\right] = \Pr\left[\hat{s} = s_t, \hat{a} = a_t \mid \tilde{\pi}_z, \mathcal{M}\right], \tag{39}$$

for all $t, \hat{s}, \hat{a}$, and

$$V_{\mathcal{M}}(\pi_z^{\mathrm{hist}}) = V_{\mathcal{M}}(\tilde{\pi}_z). \tag{40}$$

To complete the proof, we show that $\tilde{\pi}_z = \pi_z$. By Equation 37 we have

$$\Pr\left[\hat{s} = s_t, \hat{a} = a_t \mid \pi_z^{\mathrm{hist}}, \mathcal{M}\right] = \Pr\left[\hat{s} = s_t, \hat{a} = a_t \mid z, \mathcal{D}\right]. \tag{41}$$

Thus, for any $t, \hat{s}, \hat{a}$ we have

$$\tilde{\pi}_z(\hat{a} = a_t|\hat{s} = s_t) = \frac{\Pr\left[\hat{s} = s_t, \hat{a} = a_t \mid \tilde{\pi}_z, \mathcal{M}\right]}{\Pr\left[\hat{s} = s_t \mid \tilde{\pi}_z, \mathcal{M}\right]} \tag{42}$$

$$= \frac{\Pr\left[\hat{s} = s_t, \hat{a} = a_t \mid \pi_z^{\mathrm{hist}}, \mathcal{M}\right]}{\Pr\left[\hat{s} = s_t \mid \pi_z^{\mathrm{hist}}, \mathcal{M}\right]} \tag{43}$$

$$= \frac{\Pr\left[\hat{s} = s_t, \hat{a} = a_t \mid z, \mathcal{D}\right]}{\Pr\left[\hat{s} = s_t \mid z, \mathcal{D}\right]} \tag{44}$$

$$= \pi_z(\hat{a} = a_t|\hat{s} = s_t), \tag{45}$$

where the first equality is Bayes' rule, the second equality is due to Equation 39, the third equality is due to Equation 41, and last equality is by definition of $\pi_z$ (Assumption 6).

Before continuing to the main proof, we present the following analogue to Lemma 8:

**Lemma 11.** *Suppose DoC yields $q$ satisfying the MI constraints:*

$$\mathrm{MI}(r_t; z|s_t, a_t) = \mathrm{MI}(s_{t+1}; z|s_t, a_t) = 0, \tag{46}$$

*for all $s_t, a_t$ with $\Pr[s_t, a_t|\mathcal{D}] > 0$. Then under Assumptions 2 and 5,*

$$\Pr\left[\hat{r} = r_t \mid s_t, a_t, z, \mathcal{D}\right] = \mathcal{R}(\hat{r}_t \mid s_t, a_t), \tag{47}$$
$$\Pr\left[\hat{s}_{t+1} = s_{t+1} \mid s_t, a_t, z, \mathcal{D}\right] = \mathcal{T}(\hat{s}_{t+1} \mid s_t, a_t), \tag{48}$$

*for all $s_t, a_t, z$ and $\hat{r}, \hat{s}_{t+1}$, as long as $\Pr[s_t, a_t, z|\mathcal{D}] > 0$.*

*Proof.* The proof is analogous to the proof of Lemma 8.

## B.1 THEOREM PROOF

We can now tackle the proof of Theorem 7. To do so, we start by interpreting the episodes $\tau$ in the training data $\mathcal{D}$ as coming from a modified Markovian environment $\mathcal{M}^\dagger$. Specifically, we define $\mathcal{M}^\dagger$ as an environment with the same state space as $\mathcal{M}$ but with an action space consisting of tuples $(a, r, s')$, where $a$ is an action from the action space of $\mathcal{M}$, $r$ is a scalar, and $s'$ is a state from the state space of $\mathcal{M}$. We define the reward and transition functions of $\mathcal{M}^\dagger$ to be deterministic, so that the reward and next state associated with $(a, r, s')$ is $r$ and $s'$, respectively. This way, we may interpret any episode $\tau = (s_t, a_t, r_t)_{t=0}^H$ in $\mathcal{M}$ as an episode

$$\tau^\dagger = (s_t, (a_t, r_t, s_{t+1}), r_t)_{t=0}^H \tag{49}$$

in the modified environment $\mathcal{M}^\dagger$. Denoting $\mathcal{D}^\dagger$ as the training data distribution when interpreted in this way, we note that the MI constraints of Lemma 10 hold, since rewards and transitions are deterministic. Thus, the policy $\pi^\dagger$ defined as

$$\pi^\dagger((\hat{a}, \hat{r}, \hat{s}')|s_t, z) = \Pr[(\hat{a}, \hat{r}, \hat{s}') = (a_t, r_t, s_{t+1})|s_t, z, \mathcal{D}^\dagger] \tag{50}$$

satisfies

$$V(z) = V_{\mathcal{M}^\dagger}(\pi_z^\dagger). \tag{51}$$

It is left to show that $V_{\mathcal{M}^\dagger}(\pi_z^\dagger) = V_{\mathcal{M}}(\pi_z)$. To do so, consider an episode $\tau^\dagger \sim \Pr[\cdot|\pi_z^\dagger, \mathcal{M}^\dagger]$. For any single-step transition in this episode,

$$(s_t, (a_t, r_t, s_{t+1}), r_t, s_{t+1}), \tag{52}$$

we have, by definition of $\pi_z^\dagger$,

$$\Pr[\hat{a} = a_t|s_t, \pi_z^\dagger] = \Pr[\hat{a} = a_t|s_t, z, \mathcal{D}^\dagger] = \pi_z(\hat{a}|s_t). \tag{53}$$

In a similar vein, by definition of $\pi_z^\dagger$ and Lemma 11 we have,

$$\Pr[\hat{r} = r_t|s_t, a_t, \pi_z^\dagger] = \Pr[\hat{r} = r_t|s_t, a_t, z, \mathcal{D}^\dagger] = \mathcal{R}(\hat{r}|s_t, a_t), \tag{54}$$

$$\Pr[\hat{s}_{t+1} = s_{t+1}|s_t, a_t, \pi_z^\dagger] = \Pr[\hat{s}_{t+1} = s_{t+1}|s_t, a_t, z, \mathcal{D}^\dagger] = \mathcal{T}(\hat{s}_{t+1}|s_t, a_t). \tag{55}$$

Thus, any $\tau^\dagger = (s_t, (a_t, r_t, s_{t+1}), r_t)_{t=0}^H$ sampled from $\pi_z^\dagger, \mathcal{M}^\dagger$ can be mapped back to a $\tau = (s_t, a_t, r_t)_{t=0}^H$ in the original environment $\mathcal{M}$, where $\Pr[\tau^\dagger|\pi_z^\dagger, \mathcal{M}^\dagger] = \Pr[\tau|\pi_z, \mathcal{M}]$. It is clear that $R(\tau^\dagger) = R(\tau)$, and so we immediately have

$$V_{\mathcal{M}^\dagger}(\pi_z^\dagger) = V_{\mathcal{M}}(\pi_z), \tag{56}$$

as desired.

## C  INVALIDITY OF ALTERNATIVE CONSISTENCY FRAMEWORKS

Paster et al. (2022) propose a similar but distinct notion of consistency compared to ours (i.e., Definition 1), and claim that it can be achieved with stationary policies in Markovian environments. In this section, we show that this is, in fact, false, supporting the benefits of our framework. We begin by rephrasing Theorem 2.1 of Paster et al. (2022) using our own notation:

**(Incorrect) Theorem 2.1 of Paster et al. (2022).** *Suppose $\mathcal{M}$ is Markovian and $\mathcal{D}, q$ are given such that*

$$\Pr[\hat{s}_{t+1} = s_t \mid s_t, a_t, z, \mathcal{D}] = \Pr[\hat{s}_{t+1} = s_t \mid s_t, a_t, \mathcal{D}], \tag{57}$$

*for all $s_t, a_t, z, \hat{s}_{t+1}$ with $\Pr[s_t, a_t, z|q, \mathcal{D}] > 0$ and define a Markov policy $\pi$ as*

$$\pi(\hat{a}|s_t, z) = \Pr[\hat{a} = a_t|s_t, z, \mathcal{D}]. \tag{58}$$

*Then for any $z$ with $\Pr[z|q, \mathcal{D}] > 0$ and any $\tau$,*

$$\Pr[\tau \mid \pi_z, \mathcal{M}] > 0 \text{ if and only if } \Pr[\tau \mid z, \mathcal{D}] > 0. \tag{59}$$

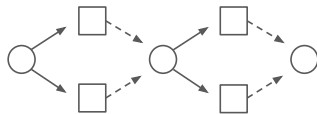

Figure 5: Deterministic environment used in the counter-example described in Appendix C. Circles represent states and squares represent actions; solid arrows represent choice of actions and dashed arrows represent environment dynamics.

**Counter-example.** A simple counter-example may be constructed by considering the Markovian environment displayed in Figure 5. The environment has three states. The first state gives a choice of two actions ($a_0 \in \{0, 1\}$), and each action deterministically transitions to the same second state. The second state again provides a choice of two actions ($a_1 \in \{0, 1\}$), and each of these again deterministically transitions to the same terminal state. Thus, episodes in this environment are uniquely determined by choice of $a_0, a_1$. There are four unique episodes:

$$\tau_0 = \langle a_0 = 0, a_1 = 0 \rangle, \tag{60}$$
$$\tau_1 = \langle a_0 = 1, a_1 = 1 \rangle, \tag{61}$$
$$\tau_2 = \langle a_0 = 0, a_1 = 1 \rangle, \tag{62}$$
$$\tau_3 = \langle a_0 = 1, a_1 = 0 \rangle. \tag{63}$$

We now construct $q$ as a deterministic function, clustering these four trajectories into two distinct $z$:

$$z_0 = q(\tau_0) = q(\tau_1), \tag{64}$$
$$z_1 = q(\tau_2) = q(\tau_3). \tag{65}$$

Suppose $\mathcal{D}$ includes $\tau_0, \tau_1, \tau_2, \tau_3$ with equal probability. Since the environment is deterministic, the conditions of Theorem 2.1 in Paster et al. (2022) are trivially satisfied. Learning a policy $\pi$ with respect to $z_0$ yields

$$\pi(\cdot|s_0, z_0) = [0.5, 0.5], \tag{66}$$
$$\pi(\cdot|s_1, z_0) = [0.5, 0.5]. \tag{67}$$

However, it is clear that interacting with $\pi(\cdot|\cdot, z_0)$ in the environment will lead to $\tau_2, \tau_3$ with non-zero probability, while $\tau_2, \tau_3$ are never associated with $z_0$ in the data $\mathcal{D}$. Contradiction.

## D PSEUDOCODE FOR DOC TRAINING

---

**Algorithm 2** Training with Dichotomy of Control

---

**Inputs** Offline dataset $\mathcal{D} = \{\tau^{(m)}\}_{m=1}^M$ where $\tau^{(m)} = (s_t^{(m)}, a_t^{(m)}, r_t^{(m)})_{t=0}^H$ with initial states $\{s_0^{(m)}\}_{m=1}^M$ and initial return-to-go values $\{R^{(m)}\}_{m=1}^M$, a parametrized distribution $q_\phi(\cdot)$, a policy $\pi_{\theta_1}(\cdot, \cdot)$, a value function $V_{\theta_2}(\cdot)$, a prior $p_\psi(\cdot)$, an energy function $f_w(\cdot)$, a fixed distribution $\rho(r, s')$, learning rates $\eta$, and training batch size $B$.
**while** training has not converged **do**
    Sample batch $\{(\tau = (s_t, a_t, r_t)_{t=0}^H)^{(m)}\}_{m=1}^B$ from $\mathcal{D}$, for $m = 1, \ldots, B$.
    Sample $z$ from $q_\phi(\tau)$ with reparametrization.
    Compute $\mathcal{L}_{\text{DoC}} + \mathcal{L}_{aux}$ according to Equation 8 and Equation 9.
    Update $\phi \leftarrow \phi - \eta \nabla_\phi \hat{\mathcal{L}}$, $\psi \leftarrow \psi - \eta \nabla_\psi \text{stopgrad}(\hat{\mathcal{L}}, \phi)$, $w \leftarrow w + \eta \nabla_w \hat{\mathcal{L}}$, $\theta_1 \leftarrow \theta_1 - \eta \nabla_{\theta_1} \hat{\mathcal{L}}$,
    $\theta_2 \leftarrow \theta_2 - \eta \nabla_{\theta_1} \hat{\mathcal{L}}$.
**return** $\pi_{\theta_1}(\cdot, \cdot), V_{\theta_2}(\cdot), p_\psi(\cdot)$

---

# E  EXPERIMENT DETAILS

## E.1  HYPERPARAMETERS.

We use the same hyperparameters as the publically available Decision Transformer (Chen et al., 2021) implementation. For VAE, we additionally learn a future and a prior both parametrized the same as the policy using transformers with context length 20. All models are trained on NVIDIA GPU P100.

Table 1: Hyperparameters of Decision Transformer, future-conditioned VAE, and Dichotomy of Control.

| Hyperparameter | Value |
| --- | --- |
| Number of layers | 3 |
| Number of attention heads | 1 |
| Embedding dimension | 128 |
| Latent future dimension | 128 |
| Nonlinearity function | ReLU |
| Batch size | 64 |
| Context length $K$ | 20 FrozenLake, HalfCheetah, Hopper, Humanoid, AntMaze |
| | 5 Reacher |
| Future length $K_f$ | Same as context length $K$ |
| Return-to-go conditioning for DT | 1 FrozenLake |
| | 6000 HalfCheetah |
| | 3600 Hopper |
| | 5000 Humanoid |
| | 50 Reacher |
| | 1 AntMaze |
| Dropout | 0.1 |
| Learning rate | $10^{-4}$ |
| Grad norm clip | 0.25 |
| Weight decay | $10^{-4}$ |
| Learning rate decay | Linear warmup for first $10^5$ training steps |
| $\beta$ coefficient | 1.0 for DoC, Best of 0.1, 1.0, 10 for VAE |

## E.2  DETAILS OF THE OFFLINE DATASETS

**FrozenLake.**  We train a DQN (Mnih et al., 2013) policy for 100k steps in the original 4x4 Frozen-Lake Gym environment with stochasticity level $p = \frac{1}{3}$. We then modify $p$ to simulate environments of different stochasticity levels, while collecting 100 trajectories of maximum length 100 at each level using the trained DQN agent with probability $\epsilon$ of selecting a random action as opposed to the action output by the DQN agent to emulate offline data with different quality.

**Gym MuJoCo.**  We train SAC (Haarnoja et al., 2018) policies on the original set of Gym MuJoCo environments for 100M steps. To simulate stochasticity in these environments, we modify the original Gym MuJoCo environments by introducing noise to the actions before inputting the action to the physics simulator to compute rewards and next states. The noise has 0 mean and standard deviation of the form $(1 - e^{-0.01 \cdot t}) \cdot \sin(t) \cdot \sigma$ where $t$ is the step number and $\sigma \in [0, 1]$. We then collect 1000 trajectories of 1000 steps each for all environments except for Reacher (which has 50 steps in each trajectory) in the stochastic version of the environment using the SAC policy to acquire the offline dataset for training.

**AntMaze.**  For the AntMaze task, we use the AntMaze dataset from D4RL (Fu et al., 2020), which contains 1000 trajectories of 1000 steps each. We add gaussian noise with standard deviation 0.1 to the rewards in the dataset uniformly with probability 0.1 to both the offline dataset and during environment evaluation to simulate stochastic rewards from the environment.

# F    ADDITIONAL RESULTS

## F.1    FROZENLAKE WITH DIFFERENT OFFLINE DATASET QUALITY.

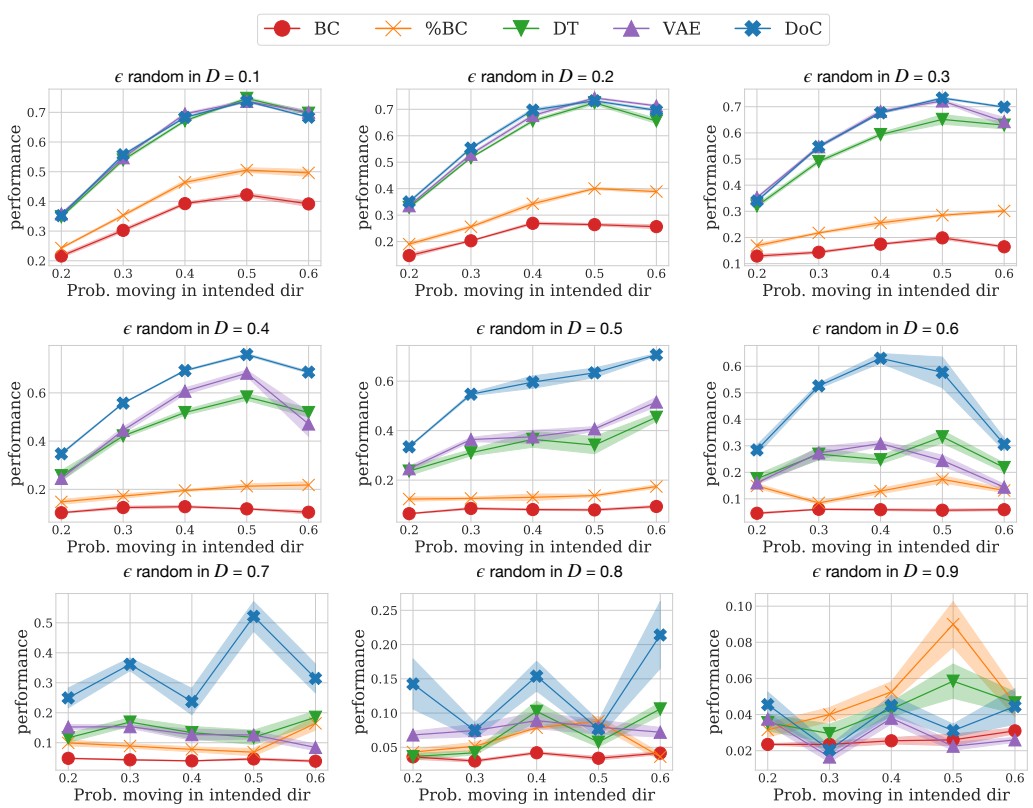

Figure 6: Average performance (across 5 seeds) of DoC and baselines on FrozenLake with different levels of stochasticity ($p$) and offline dataset quality ($\epsilon$). DoC outperforms DT and future VAE with bigger gains the offline data is less optimal.

## F.2    IMPROVEMENT OF DOC OVER RVS

To test the effect of applying the MI constraint to other future-conditioned supervised learning baselines, we evaluate RvS parametrized by MLP policies (Emmons et al., 2021) with VAE and DoC modifications. In general, MLP parametrization performs worse than transformer parametrization, but DoC is still able to provide significant benefit over vanilla RvS.

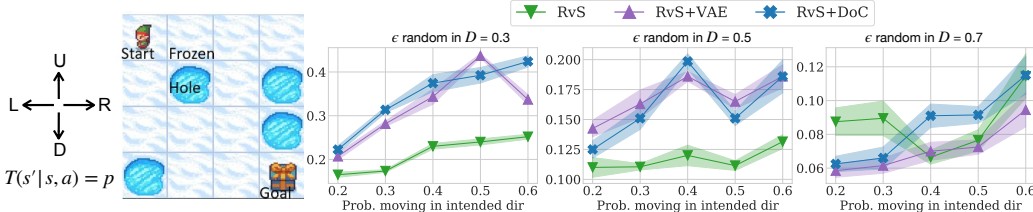

Figure 7: Average performance (across 5 seeds) of DoC and baselines on FrozenLake with different levels of stochasticity ($p$) and offline dataset quality ($\epsilon$). DoC outperforms RvS and future VAE with bigger gains the offline data is less optimal.

## G    ADDITIONAL ABLATIONS

### G.1    VAE WITH STOP GRADIENT

One difference between DoC and VAE is whether there is a stop gradient operation on the posterior $q(z|\tau)$ when minimizing the KL-divergence between $q(z|\tau)$ and the prior $p(z|s_0)$. We conduct the ablation below in Figure 8 where we also apply stop gradient to VAE, and observe that VAE's performance drops significantly.

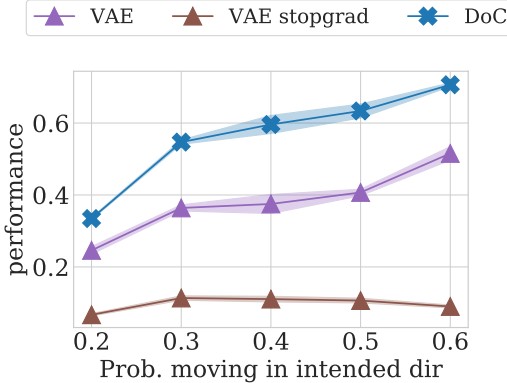

Figure 8: Average performance (across 5 seeds) of DoC and baselines on FrozenLake with different levels of stochasticity ($p$) and offline dataset quality ($\epsilon$). DoC outperforms RvS and future VAE with bigger gains the offline data is less optimal.

## G.2 DoC with different regularization strength ($\beta$)

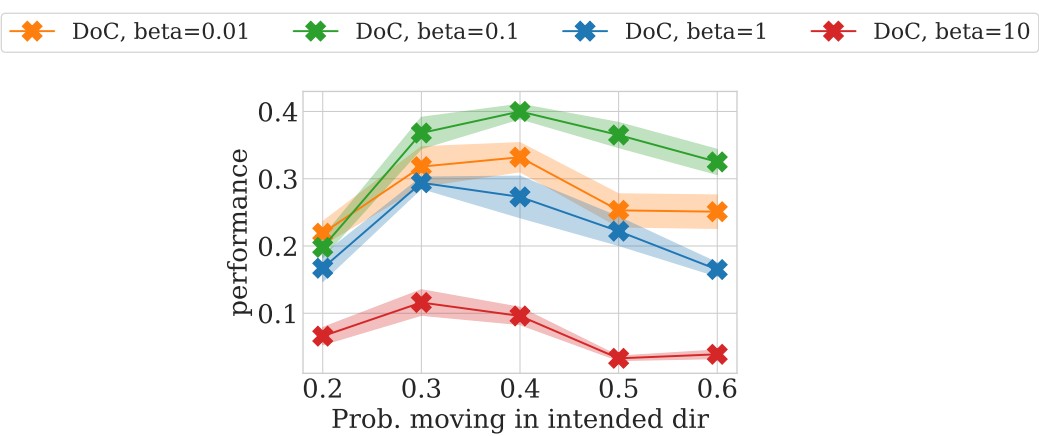

Figure 9: Average performance (across 5 seeds) of DoC with different regularization strength ($\beta$). The effect of $\beta$ is more pronounced when the dataset is highly optimal (e.g., $\epsilon$ random in $D = 0.7$), for which we found a smaller $\beta$ (e.g., 0.1) to generally perform better.

## G.3 DoC with different number of future samples ($K$)

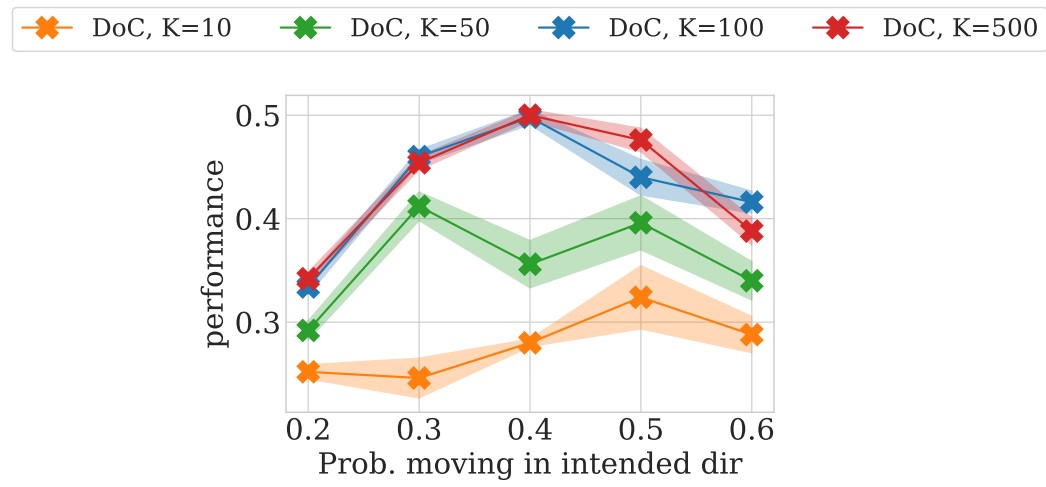

Figure 10: Average performance (across 5 seeds) of DoC with different number of samples during inference ($K$). We found that higher number of samples leads to better performance as we expect, and the gain beyound 100 samples is negligible.

