# OpenReview forum: "Dichotomy of Control: Separating What You Can Control from What You Cannot"
_ICLR.cc/2023/Conference — ICLR 2023 notable top 5%_

### Official Review · Reviewer_eVkG · 2022-10-24

**Confidence:** 4
**Correctness:** 4
**Technical Novelty And Significance:** 4
**Empirical Novelty And Significance:** 3
**Recommendation:** 8

**Clarity, Quality, Novelty And Reproducibility:**

The writing is very clear and foundations are well explained. However,at the key contribution: the addition of constraints in the objective using mutual information and the derivation of the objective, here the authors could have made the steps and key assumption more clearer.

**Strength And Weaknesses:**

Pros
- The paper is written very well and good to follow
- The paper is focused on a relevant question in the field
- Concepts are well introduced and explained
- Experiments make sense and are executed nicely: testing in stochastic scenarios of different scale -- visualization, experiment repetition etc.


Cons:
My main critique is around the point of universality:  I feel a bit unsure about how universal the approach is for different forms of stochasticity in the environment and how well it would perform if no stochasticity would be present:

 1. I feel a bit unsure about Eq. (7) (especially in continuous cases): to calculate the mutual information between the reward r_t and z (and s_{t+1}) you say you "we set ρ to be the marginal distribution of rewards in the dataset" and that is has to be a  "fixed sampling distribution of rewards"  But this sounds like you will have to make simplifying assumptions here about the stochasticity of the s_t and r_t:
    - are you limited to Gaussian and empirical (histogram) distributions? What happens if the environment is multi-modal or long-tail in its state or reward transitions?
    -  do you learn the parameters of the conditional distribution \omega?
    -  what happens if the environment would be stochastic in some regions and deterministic in others?
    -  Looking at the experiment, I noticed the algorithm is only tested in situations where randomness is either Bernoulli (6.1, 6.2) or Gaussian (6.3).


2. The authors evaluate their method solely on stochastic tasks (or deterministic tasks, like MuJoCo but made stochastic). I do not know how well the method performs under standard (deterministic) settings. E.g. perhaps it will behave a bit more conservative because it assumes some form of stochastic behavior







**Summary Of The Paper:**

The paper is in the field of "Future- or return-conditioned supervised learning". Here the main idea is to learn, from a batch of data, a policy that is conditioned on a latent variable z, which is often the return of a given trajectory.  However, in stochastic environments, the return is often not driven by the performance of the actions executed,  but by the inherent stochasticity of the environment.

Here, the authors introduce a novel objective function, by penalizing the mutual information between the latent variable and the (environment-driven) stochastic rewards. (The method is (loosely) inspired by the stoic concept of dichotomy of control, which they used to name the method and the paper).

The authors  test this method on a set of stochastic benchmarks  (often deterministic, but adjusted to be stochastic) and show better stability and performance

**Summary Of The Review:**

The methodology is novel and its performance is empirically shown.   I have some doubts regarding the universality of the approach, which may be cleared up during the rebuttal and subsequent discussions.

Therefore, for now I vote for acceptance with some reservations regarding empirical significance.

---

> ### Author Response · Authors · 2022-11-10
> **Response to Reviewer eVkG**
>
> Thank you for the highly positive feedback. Please let us know if the following addresses your concerns.
>
> > Simplifying assumptions here about the stochasticity of the $s_t$ and $r_t$
>
> We do not make any simplifying assumption (e.g., Gaussian) about stochasticity of $s_t$ and $r_t$. In contrastive learning of energy-based models, one can set $\rho$ to be anything, and the energy function can capture any multi-modal distributions in the exponential family form [1][2]. Since the choice of $\rho$ is flexible, we choose a  $\rho$ that is more convenient for implementation. Such a choice is standard in implementations of contrastive learning of energy-based models.
>
> > Do you learn the parameters of the conditional distribution \omega?
>
> $\omega$ is only used for derivation and is completely parameterized by the energy function $f$. Thus, during optimization only the parameters of $f$ are learned.
>
> > What happens if the environment would be stochastic in some regions and deterministic in others?
>
> DoC will treat deterministic regions as is (not being overly conservative) as each deterministic region will simply be encoded as its own latent variable value.
>
> > How well the method performs under standard (deterministic) settings.
>
> We have empirically tested DoC on deterministic environments and it performed similar to DT. Theoretically, if $z$ is set to the future returns-to-go, DoC will reduce exactly to DT.
>
> [1] LeCun, Yann, Energy-Based Self-Supervised Learning, http://helper.ipam.ucla.edu/publications/mlpws4/mlpws4_15927.pdf
>
> [2] Gutmann, Michael, and Aapo Hyvärinen. "Noise-contrastive estimation: A new estimation principle for unnormalized statistical models." Proceedings of the thirteenth international conference on artificial intelligence and statistics. JMLR Workshop and Conference Proceedings, 2010.

---

### Official Review · Reviewer_aa3o · 2022-10-25

**Confidence:** 4
**Clarity, Quality, Novelty And Reproducibility:** Mentioned above
**Correctness:** 3
**Technical Novelty And Significance:** 3
**Empirical Novelty And Significance:** 3
**Recommendation:** 6

**Strength And Weaknesses:**

Strength
- Empirical performance on environments with high stochasticity is strong compared to other future-conditioned supervised learning baselines.
- The paper presents various examples and experiments to backup their claims.

Weaknesses
- DoC inference requires sampling latents from the learned prior and choosing the latent with the highest (estimated) value. I worry this process adds randomness to the policy's performance and thus is unstable. Also, if there exists only few expert quality trajectories on the dataset and other trajectories are low quality, much more sampling will be required to include a latent with good returns. In addition, the paper does not discuss the protocol for setting K (number of samples hyperparameter).
- The authors did not provide the source code, which makes the credibility of the paper questionable.
- The paper lacks results on hyperparameter sensitivity. For example, how does the return curve change if we change \beta and K?

Questions
- On Gym MuJoCo, why is time-correlated noise used instead of simple Gaussian noise?
- Can the authors provide results on applying the proposed regularizer to other future-conditioned supervised learning baselines, for example RvS [1]?

[1] Emmons et al., RvS: What is Essential for Offline RL via Supervised Learning?, ICLR 2022.

######## Post-rebuttal comment ########

The updated manuscript addresses my concerns mentioned above. Thus, I am raising my score.

**Summary Of The Paper:**

This paper proposes a new regularizer for future-conditioned supervised learning that specifically prevents the learning procedure from being biased by lucky rollouts (in terms of environment stochasticity). On highly stochastic environments, the proposed method achieves higher performance compared to the baselines.

**Summary Of The Review:**

The paper claims that current future-conditioned supervised RL methods lack the ability to distinguish environment randomness from policy randomness and introduces a new regularizer to address this. While the proposed regularizer seems to help boost performance on highly stochastic environments, more experiments and clarifications of the proposed method is required.

---

> ### Author Response · Authors · 2022-11-10
> **Response to Reviewer aa3o**
>
> Thank you for the positive review on our experimental results. Per your suggestion, we include additional ablations and experiments in the Appendix. Please let us know if the following addresses your concerns.
>
> > This process adds randomness to the policy's performance and thus is unstable.
>
> Our inference procedure is heavily inspired by highly-performant Decision Transformer implementations [1], which also sample a bunch of future returns at each inference step and pick the higher (highest) one to condition on. DoC does not introduce more randomness compared to this implementation of DT. Future work may benefit from learning a high-level policy to select latent variable values directly.
>
> > The authors did not provide the source code.
>
> We have updated the supplementary material with code and instructions for reproducing the experimental results.
>
> > Effect of $\beta$ and $K$.
>
> We used $\beta=1.0$ and $K=100$ in our original paper, which we found to generally perform well. We include ablation studies of $\beta$ in [Figure 10](https://i.ibb.co/kcFf1sJ/lake-beta.jpg) and $K$ in [Figure 11](https://i.ibb.co/zZ0ZKJ4/lake-K.jpg) in the updated Appendix. In particular, we found that when the offline dataset is highly suboptimal, a smaller $\beta$ value tends to work better. Higher number of samples generally leads to better performance, though beyond 100 samples, the performance remained roughly the same.
>
> > On Gym MuJoCo, why is time-correlated noise used instead of simple Gaussian noise?
>
> Experiments using simple Gaussian noise were not able to distinguish between different methods. In long-horizon (length 1000) MuJoCo environments, the effect of simple Gaussian noise may be “averaged” over the large number of future timesteps. Thus, we found time-correlated noise to exhibit much more multi-modal/long-tail behavior, which makes the stochasticity in the environment more sophisticated and hence difference between methods more pronounced.
>
> > Can the authors provide results on applying the proposed regularizer to other future-conditioned supervised learning baselines, for example RvS [1]?
>
> We include additional results on applying DoC to RvS parametrized using MLP policies in [Figure 7](https://i.ibb.co/sghgPSy/rvs-lake.jpg) of the updated appendix. In general, MLP parametrization with RvS performs worse than transformer parametrization with DT, but DoC is still able to provide significant benefit over vanilla RvS.
>
> [1] Lee, Kuang-Huei, et al. "Multi-Game Decision Transformers." arXiv preprint arXiv:2205.15241 (2022).

---

> > ### Comment · Reviewer_aa3o · 2022-11-16
> > **Response to Authors**
> >
> > Thanks for the detailed response and sorry for the late reply.
> >
> > I appreciate the authors' effort on providing the source code and running new experiments. Regarding the inference procedure, I think it is unfair to compare the complexity (or randomness) of the proposed method with [1] as the experiment section of the paper only includes the vanilla DT. Also, if your inference procedure is heavily inspired by [1], it would be better to note it on explicitly on the paper.
> >
> > Minor comments: The result in Figure 10 is interesting in that it shows larger K can sometimes lead to worse performance (K: 100 -> 500, Prob. moving in intended dir = 0.6). Could it be because the chosen latent z* can overfit to the value function? Regarding this matter, what happens if change the inference procedure from random sampling and choosing the best latent to directly optimizing the latent with regard to the value function? I would understand if the authors do not reply to these comments as my response was late, but I think including these discussions will make the future version of the paper more complete.

---

> > > ### Author Response · Authors · 2022-11-16
> > > **Response to Reviewer aa3o**
> > >
> > > Thank you for the thoughtful follow-up. DoC does incur additional computational overheads compared to vanilla DT. We have updated the paper (last sentence in Section 4) to reflect this limitation of DoC. We have also updated the paper (blue text in Section 6) to reflect that DoC’s inference procedure is heavily inspired by [1].
> > >
> > > Overfitting to the value function could be one explanation for why K = 500 is (though only slightly) worse than K = 100 when probability of moving in the intended direction equals 0.6. Indeed, the inference-time sampling of z may choose a z that the value model never saw during training, and thus lead to unintended behavior. In such situations, one may want to mitigate this phenomenon by applying a regularizer to the choice of z to penalize out-of-distribution latents.
> > >
> > > Please let us know if you have any other concerns.
> > >
> > > [1] Lee, Kuang-Huei, et al. "Multi-Game Decision Transformers." arXiv preprint arXiv:2205.15241 (2022).

---

> > > > ### Author Response · Authors · 2022-12-02
> > > > **Follow-up to Reviewer aa3o**
> > > >
> > > > Dear Reviewer, we would like to ask if your concerns have been addressed by our comments and additional results, or if there were any other issues we can address. Please let us know, and thank you for your time.

---

> > > > > ### Comment · Reviewer_aa3o · 2022-12-05
> > > > > **Response to Authors**
> > > > >
> > > > > Thank you for the response.
> > > > >
> > > > > The updated manuscript addresses my concerns and I am raising my score.

---

### Official Review · Reviewer_LLnY · 2022-10-30

**Confidence:** 3
**Correctness:** 3
**Technical Novelty And Significance:** 4
**Empirical Novelty And Significance:** 3
**Recommendation:** 8

**Clarity, Quality, Novelty And Reproducibility:**

The paper elaborates most of the conclusions clearly and presents a new and novel view of the stochasticity in the RL environment, but with few confusions left, leaving out derivation details without any reference to the appendix. Most of the notations, theories, and assumptions are accessible. I have listed a few points I am confused about below:
1. How do you convert the problem of minimizing the mutual information to maximizing the same equations derived from mutual information?
2. How to calculate the corresponding value function for a give z during inference is ambiguous


**Strength And Weaknesses:**

Strength:
1. The paper has clearly pointed out the problem sets they are trying to improve on. And the way they came to their choices on mutual information sounds natural.
2. Most of the notations and the derivations are clear to follow
3. The experiment setup and resulting figures are clear

Weakness:
1. Some part of the derivation mentioned in the next section is confusing to me
2. Some of the phrasings are very abstract and can be more rigorous, such as "By only capturing the controllable factors in the latent variable, DoC can maximize over each action step without also attempting to maximize environment transitions as shown in Figure 1".

**Summary Of The Paper:**

After reviewing the limitations of the current RCSL methods in capturing reward stochasticity, this paper has proposed a new Learning algorithm DoC that solves the control problem in a stochastic environment by separating what is controllable from what is not. The former is formulated as the actions at hand at each timestep, while the latter is interpreted as the stochastic environment transition probabilities. They have learned a new latent variable to represent future information and aid inference and included mutual information constraints in the optimization problem. Experiments comparing DoC with mainly RCSL/DT and future VAE methods proved the novelty of the paper in terms of the recovery of the optimal policies.

**Summary Of The Review:**

Regardless of the derivation confusion, it causes me, it is a novel and original paper to me from both points of view of optimization goal formulation and empirical results. I would consider this an inspiring direction to which I would like to contribute.

---

> ### Author Response · Authors · 2022-11-10
> **Response to Reviewer LLnY**
>
> Thank you for the positive review. We have updated the text (highlighted in blue) according to your feedback.
>
> > Some of the phrasings are very abstract and can be more rigorous
>
> We agree that some phrasing in the introduction was vague. We have updated that sentence to "DoC only captures information from the controllable actions and avoids capturing information from the uncontrollable environment transitions in the latent variable so that maximization only happens with respect to the controllable actions."
>
> > How do you convert the problem of minimizing the mutual information to maximizing the same equations derived from mutual information?
>
> Minimizing the mutual information becomes maximizing over the energy-based function $f$ through the Lagrangian of the constrained optimization problem in Equation 4, which we derive in Section 4.2. Similar derivations have also been employed by prior work such as [1].
>
> > How to calculate the corresponding value function for a give z during inference is ambiguous
>
> During training, we learn a value function as an auxiliary objective in Equation 9. This objective is a simple regression loss. The learned value function is then used during inference to select the sampled latent variable values.
>
> [1] Belghazi, Mohamed Ishmael, et al. "Mine: mutual information neural estimation." arXiv preprint arXiv:1801.04062 (2018).

---

### Official Review · Reviewer_kJLT · 2022-11-04

**Confidence:** 4
**Correctness:** 2
**Technical Novelty And Significance:** 3
**Empirical Novelty And Significance:** 2
**Recommendation:** 6

**Clarity, Quality, Novelty And Reproducibility:**

The paper is clearly written with good quality and novelty. However, certain experiments are needed to justify the novelty.

**Strength And Weaknesses:**

## Strengths
- The paper identifies an important problem and solution for RCSL with inconsistent conditioning, especially in stochastic environments.
- The paper is clearly written with all the necessary details and flow to understand everything.
- A good set of simple environments are chosen, including a didactic Bernoulli Bandit environment and MuJoCo benchmarks.
- The paper has theoretical results on the consistency of their proposed method.


## Weaknesses
- **Missing crucial comparisons**
    + There are two key differences of DoC from prior approaches (i.e., VAE) using latent embedding of future to mitigate inconsistency: (a) Mutual Information (MI) constraint and (b) Inference from a learned prior and value function. The importance of these two components must be ablated on all the environments. Specifically, the following comparisons can be added:
        * VAE + Inference with learned prior (a separate copy with stopgrad) and value function
        * DoC w/o MI constraint (= DoC - a)
        * DoC + conditioning on highest return (= DoC - b)
- **Comparison against VAE**
    + Since future-VAE also regularizes the z to the learnable prior conditioned on the past, this ensures that the latent z is incentivized not to use future information. Therefore, DoC's key benefit must come from utilizing the controllable part of the future in z while ignoring the environment transitions. Why is it expected that encoding the controllable information in z will improve the performance? What are example environments with this property?
    + The difference between the ideology of VAE and DoC being seemingly small is reinforced by the results on the mujoco environments, where the empirical performance of VAE and DoC are pretty similar. Even on reacher, where KL beta is not tuned well, VAE first reachers almost the optimal performance before falling. Therefore, it is not clear that DoC is necessarily better than VAE.
        * Is more stochasticity in mujoco environments expected to increase the difference in performance between DoC and VAE?
        * Is any other experiment (quantitative or qualitative) possible to show that the importance of encoding controllable future information in z is helpful?

**Summary Of The Paper:**

This paper aims to address the issue of inconsistency in return-conditioned supervised learning (RCSL), such as Decision Transformers. Specifically, when RCSL in highly stochastic environments is conditioned on the highest dataset return, the resulting expected return could be lower as the environment randomness is not under the agent's control. To this end, this paper proposes to capture only the agent's controllable factors and minimize the learning dependence on future rewards and transitions — which are environment characteristics. Under reasonable assumptions, the paper proves that their latent variable approach with Mutual Information constraints leads to policies consistent with their conditioning input reward.

**Summary Of The Review:**

I like the paper's writing, problem, and proposed approach. However, it is missing some crucial justification as to why it is expected to be better than the best baseline (VAE), which prior work has proposed to mitigate inconsistency. I would happily reconsider my rating if the above issues are addressed, resulting in a clear demonstration of improvement over baselines and ablations.

----
[Post-rebuttal]
The author response addressed most of my concerns, except "DoC does not convincingly outperform VAE in control tasks." I have increased my score to reflect this.

---

> ### Author Response · Authors · 2022-11-10
> **Response to Reviewer kJLT**
>
> Thank you for the detailed review. We’re glad to hear that you found our paper to be of good quality and novelty. We address the concerns regarding the difference between DoC and VAE below. Please let us know if these comments adequately address all your concerns or if you have any further questions.
>
> > It is not clear that DoC is necessarily better than VAE.
>
> Theoretically, VAE has essentially no guarantees for future-conditioned supervised learning in the literature, whereas DoC provides consistency guarantees, which is a major contribution of this work. We have updated the related work (blue text) to further clarify this point.
>
> > Ablations on VAE’s learned prior and MI constraints.
>
> We want to clarify that all of our implementations – both VAE and DoC – use learned prior and value functions to select actions at inference time (VAE without a value function performs poorly). The differences between VAE and DoC are (1) DoC uses the MI constraint as opposed to the ELBO during max-likelihood training, and (2) The gradient of the prior and value function does not affect DoC’s latent variable (i.e., the stop-gradient in the “DoC Inference” paragraph in Section 4.2). Following the reviewer’s suggestions, we have conducted an ablation to evaluate VAE augmented with a stop-gradient on the learned prior and value function, and observe significant degradation in VAE’s performance as shown in [Figure 8](https://i.ibb.co/YcPXzxq/lake-vae-stopgrad.jpg) in the updated Appendix. Note that this variant with stopgrad corresponds to the “DoC without MI constraint” ablation that the reviewer suggested.
>
> > Ablation on DoC + conditioning on highest return (= DoC - b)
>
> To avoid us misinterpreting you, could you clarify what ablation specifically you are interested in? Since DoC is trained to condition on the latent future $z$, it is not obvious how DoC can condition on highest return instead during inference. DoC can only condition on the $z$ that corresponds to the highest return, which is what we do in practice.
>
> > Why is it expected that encoding the controllable information in z will improve the performance? What are example environments with this property?
>
> Only encoding the controllable information in $z$ is required to achieve consistency (i.e., a conditional policy achieves the condition in the MDP). If $z$ incorporates lucky returns (e.g., from winning a lottery), the policy that’s conditioned on this lucky return will not actually achieve that return in expectation, and VAE does not resolve this issue as it naively encodes the lucky return in $z$. Environments that are highly stochastic are expected to benefit from DoC. For instance, self-driving cars involving other stochastic drivers will benefit from DoC. Financial market is another example, where a DT / VAE approach can lead to overly optimistic policies (suggesting buying lottery tickets as opposed to exchange traded funds).
>
> > Is more stochasticity in mujoco environments expected to increase the difference in performance between DoC and VAE?
>
> Yes, the benefits of DoC over VAE is more pronounced in more stochastic environments. For instance, we added time-dependent noise in MuJoCo to see an effect (as opposed to Gaussian noise, which doesn’t show a significant difference, since a lot of the randomness is “averaged” over the long horizon used in MuJoCo environments).

---

> > ### Author Response · Authors · 2022-12-02
> > **Follow-up to Reviewer kJLT**
> >
> > Dear Reviewer, we would like to ask if your concerns about DoC and VAE have been addressed by our comments and additional results. Thank you.

---

> > ### Comment · Reviewer_kJLT · 2022-12-04
> > **Most concerns addressed**
> >
> > Thank you for your response and apologies for my late response. Your rebuttal addresses most of my concers, especially with:
> > - "both VAE and DoC – use learned prior and value functions to select actions at inference time (VAE without a value function performs poorly)"
> > - "DoC can only condition on the  that corresponds to the highest return, which is what we do in practice."
> >
> > My only remaining dissatisfaction is that DoC does not convincingly outperform VAE on the chosen MuJoCo tasks. This is why I originally asked:
> > > Is any other experiment (quantitative or qualitative) possible to show that the importance of encoding controllable future information in z is helpful?
> >
> > I was hoping the tested benchmarks are such that DoC > VAE clearly, i.e. where encoding controllable future information is necessary for good performance. I understand the time-dependent noise might have been better than Gaussian noise in showing a difference, but at least to me, Figure 4 still does not show that difference clearly (unless I am missing something). Please let me know if I missed something or you have other results with a larger contrast.
> >
> > Having said that, in the light of the contributions, theoretical justifications, and the FrozenLake results, I think the paper is just above the acceptance threshold. So, I have updated my score.

---

### Decision · Program_Chairs · 2023-01-20

**Decision:**

Accept: notable-top-5%

**Justification For Why Not Higher Score:**

N/A

**Justification For Why Not Lower Score:**

The reviewers provide high scores. The reviewers point out numerous strengths of the paper, while the weaknesses are very few.

**Metareview: Summary, Strengths And Weaknesses:**


Summary:


The paper reviews the limitations of the current RCSL methods in capturing reward stochasticity. Then it proposes a new Learning algorithm DoC that solves the control problem in a stochastic environment by separating what is controllable from what is not. The former is formulated as the actions at hand at each timestep, while the latter is interpreted as the stochastic environment transition probabilities. They authors learn a new latent variable to represent future information and aid inference and included mutual information constraints in the optimization problem. Experiments comparing DoC with mainly RCSL/DT and future VAE methods proved the novelty of the paper in terms of the recovery of the optimal policies.

Strengths:

- Empirical performance on environments with high stochasticity is strong compared to other future-conditioned supervised learning baselines.
- The paper presents various examples and experiments to backup their claims.
- The paper has clearly pointed out the problem sets they are trying to improve on. And the way they came to their choices on mutual information sounds natural.
- Most of the notations and the derivations are clear to follow
- The experiment setup and resulting figures are clear
- The paper is written very well and good to follow
- The paper is focused on a relevant question in the field
- Concepts are well introduced and explained
 Experiments make sense and are executed nicely: testing in stochastic scenarios of different scale -- visualization, experiment repetition etc.
- The paper identifies an important problem and solution for RCSL with inconsistent conditioning, especially in stochastic environments.
- The paper is clearly written with all the necessary details and flow to understand everything.
- A good set of simple environments are chosen, including a didactic Bernoulli Bandit environment and MuJoCo benchmarks.
- The paper has theoretical results on the consistency of their proposed method.

Weaknesses:

- Some of the phrasings are very abstract and can be more rigorous.
-  unsure about how universal the approach is for different forms of stochasticity in the environment and how well it would perform if no stochasticity would be present:
- DoC does not convincingly outperform VAE on the chosen MuJoCo tasks

Recommendation:

All reviewers agree on acceptance. I have therefore decided to accept the paper. I recommend the authors to look at the feedback provided by the reviewers and use that to improve the paper for the camera ready version.

**Note From Pc:**

if the above contains the word "oral" or "spotlight" please see: "oral" presentation means -> notable-top-5% and "spotlight" means -> notable-top-25%. As stated in our emails, we are disassociating presentation type from AC recommendations